

# Discovery of plasma messenger RNA as novel biomarker for gastric cancer identified through bioinformatics analysis and clinical validation

Wei Cao[1,2,3,*], Dan Zhou[1,2,3,*], Weiwei Tang[4], Hanxiang An[1] and Yun Zhang[1,2,3]

[1] Department of Medical Oncology, Xiang'an Hospital of Xiamen University, School of Medicine, Xiamen University, Xiamen, Fujian, China

[2] Key Laboratory of Design and Assembly of Functional Nanostructures, Fujian Provincial Key Laboratory of Nanomaterials, Fujian Institute of Research on the Structure of Matter, Chinese Academy of Sciences, Fuzhou, China

[3] Department of Translational Medicine, Xiamen Institute of Rare Earth Materials, Chinese Academy of Sciences, Xiamen, China

[4] Department of Medical Oncology, Cancer Hospital, The First Affiliated Hospital of Xiamen University, School of Medicine, Xiamen University, Teaching Hospital of Fujian Medical University, Xiamen, Fujian, China

[*] These authors contributed equally to this work.

Corresponding authors
Hanxiang An, an_hanxiang@126.com
Yun Zhang, zhangyunfjirsm@163.com

## ABSTRACT

**Background.** Gastric cancer (GC) is the third leading cause of cancer-related death worldwide, partially due to the lack of effective screening strategies. Thus, there is a stringent need for non-invasive biomarkers to improve patient diagnostic efficiency in GC.

**Methods.** This study initially filtered messenger RNAs (mRNAs) as prospective biomarkers through bioinformatics analysis. Clinical validation was conducted using circulating mRNA in plasma from patients with GC. Relationships between expression levels of target genes and clinicopathological characteristics were calculated. Then, associations of these selected biomarkers with overall survival (OS) were analyzed using the Kaplan-Meier plotter online tool.

**Results.** Based on a comprehensive analysis of transcriptional expression profiles across 5 microarrays, top 3 over- and underexpressed mRNAs in GC were generated. Compared with normal controls, expression levels of *collagen type VI alpha 3 chain* (*COL6A3*), *serpin family H member 1* (*SERPINH1*) and *pleckstrin homology and RhoGEF domain containing G1* (*PLEKHG1*) were significantly upregulated in GC plasmas. Receiver-operating characteristic (ROC) curves on the diagnostic efficacy of plasma *COL6A3*, *SERPINH1* and *PLEKHG1* mRNAs in GC showed that the area under the ROC (AUC) was 0.720, 0.698 and 0.833, respectively. Combined, these three biomarkers showed an elevated AUC of 0.907. Interestingly, the higher *COL6A3* level was significantly correlated with lymph node metastasis and poor prognosis in GC patients. High level of *SERPINH1* mRNA expression was correlated with advanced age, poor differentiation, lower OS, and *PLEKHG1* was also associated with poor OS in GC patients.

**Conclusion.** Our results suggested that circulating *COL6A3*, *SERPINH1* and *PLEKHG1* mRNAs could be putative noninvasive biomarkers for GC diagnosis and prognosis.

## INTRODUCTION

Gastric cancer (GC) is one of the most common diagnosed cancers worldwide and the third leading cause of cancer-related mortality after lung and liver cancer (*Ferlay et al., 2015*). The carcinogenesis and progression of GC is complex, involving the alternations of multi-step and multi-genes (*Zhao et al., 2017*). The 5-year survival rate of GC diagnosed at a later stage is less than 20%, but rises to 90% for patients diagnosed at an early stage (*Stahl et al., 2017*). Although traditional biomarkers including carbohydrate antigen 19-9 (CA 19-9) and carcino-embryonic antigen (CEA) have improved clinical outcomes for GC, their sensitivity and specificity are still limited (*Ding et al., 2017*). Therefore, the identification of novel screening biomarkers may help better diagnose and improve the prognosis of GC (*Stahl et al., 2017*).

Recently, there is a growing interest in circulating messenger RNAs (mRNAs) isolated from body fluids as potential minimally/non-invasive biomarkers for cancer detection (*Kishikawa et al., 2015*; *Sole et al., 2019*). Numerous studies have reported that circulating mRNAs can be detected in GC (*Funaki et al., 1996*), melanoma (*Kopreski et al., 1999*) and nasopharyngeal carcinoma (*Lo et al., 1999*). These discoveries provide evidence for circulating mRNAs to be served as appealing non-invasive biomarker candidates in various cancers. However, few circulating mRNAs have been investigated in GC, and limited studies have compared the expression level of circulating mRNAs in plasma of GC patients with the clinicopathological characteristics.

The establishment of high-throughput molecular database such as microarray databases brings a new approach for biomarker identification. We have previously published an effective bioinformatics scheme to identify noninvasive biomarkers for lung cancer (*Zhou et al., 2017*). In the present study, we applied a similar approach to explore mRNAs circulated in plasma from patients with GC as potential biomarkers. Subsequently, the associations between these biomarkers and clinicopathological characteristics were analyzed. Finally, the relationships between these selected noninvasive biomarkers and overall survival (OS) were investigated.

## MATERIALS & METHODS

### Genome-wide expression analysis by Oncomine

Expression profiling, including 304 GC cancer samples and 174 normal controls, was obtained from Oncomine microarray database (http://www.oncomine.org) (*Rhodes et al., 2004*). In order to analyze the expression pattern of cancer vs. normal tissue mRNA, we focused on primary tumors and the following cut-offs were employed $p$-value $\leq 10^{-4}$, fold change $\geq 2$ and gene rank $\leq 10\%$. Heat maps of overexpressed and underexpressed mRNAs in GC were available for each study.

**Table 1  Clinicopathologic characteristics of gastric cancer patients and healthy subjects.**

| Clinicopathologic factors | Gastric cancer cases | Healthy subjects |
|---|---|---|
| Total | 56 | 14 |
|    Mean Age $\pm$ SD | 62.93 $\pm$ 6.17 | 43.64 $\pm$ 14.91 |
| Sex | | |
|    Male | 43 | 9 |
|    Female | 13 | 5 |
| Stage[a] | | |
|    I + II + III | 37 | |
|    IV | 18 | |
|    N/A[b] | 1 | |
| Lymph node metastasis | | |
|    <15 | 31 | |
|    $\geq$15 | 7 | |
|    N/A | 18 | |
| Differentiation | | |
|    Poor | 19 | |
|    Moderately or well-differentiated | 11 | |
|    N/A | 26 | |

**Notes.**

[a] Tumor stages were determined according to Union Internationale Contre le Cancer (UICC) criteria.

[b] N/A, not available.

## Clinical specimens

Peripheral blood samples from 56 patients with gastric adenocarcinoma were addressed before therapeutic intervention by venipuncture and processed within 2 h at the First Affiliated Hospital of Xiamen University. We also collected blood samples from 14 healthy volunteers. The healthy volunteers presented neither a history of cancer nor other diseases. All patients were pathologically diagnosed as having gastric cancer using surgical specimens and biopsies. Plasma was isolated from 4 ml blood specimens after centrifugation at 1,600$\times$ g for 10 min at 4 °C and 10,000$\times$ g for 10 min, and then stored at −80 °C until the next step. Demographic, clinical and histopathological parameters of all these cases were summarized in Table 1. All experimental protocols were approved by the Clinical Research Ethics Committee of the First Affiliated Hospital of Xiamen University. All methods were performed in accordance with the Declaration of Helsinki. Written informed consent was obtained from all human participants after complete description of the study (Fig. S1 and Table S2).

## RNA extraction

RNA was extracted from 500 µl plasma using TRIzol LS reagent (cat#10296018, Thermo Fisher Scientific Inc.) following manufacturer's instructions as previously described (*Pucciarelli et al., 2012*; *Zhang et al., 2012*). In brief, 500 µl plasma was mixed with 500 µl TRIzol reagent. After 5 min incubation at 4 °C, 500 µl chloroform was added to the mixture, and violently shaken for 30 s. The mixture was immediately centrifuged at 12,000$\times$ g for 5 min at 4 °C. The above aqueous layer was transferred into a fresh tube containing 800 µl

isopropyl alcohol. Next, the mixture was centrifuged at 12,000× g for 5 min at 4 °C and washed with 1 ml 70% ethanol for twice. Lastly, the RNA pellet was dissolved in 15 μl RNase-free water. Qubit RNA HS Assay Kit and Qubit 3.0 fluorometer (ThermoFisher Scientific Inc. cat#Q32851) were employed to quantify the concentration of RNA solution through spectrophotometry. The concentration of RNA isolated from plasma ranged was 35.24–168.18 ng/μl.

## Quantitative PCR (qPCR)

The extracted total RNA was reverse-transcription into cDNA using PrimeScript RT reagent Kit (TAKARA cat#RR047A) according to the manufacturer's protocol in triplicate. The resulting cDNA was stored at −80 °C for next PCR amplification. Primer sequences were designed through web-based version 4.1.0 of Primer 3 and were shown in Table S1. For clinical validation of the bioinformatics analysis results, qPCR was conducted by ABI ViiA 7 Real-Time PCR System (Applied Biosystems) with melting curve analysis. QPCR was carried out in triplicate at 50 °C for 2 min, denaturing at 95 °C for 5 min, followed by 40 cycles at 95 °C for 30 s, 58 °C for 1 min. *Glyceraldehyde-3-phosphate dehydrogenase (GAPDH)* was selected as an internal control. Water negative controls contained all components for the qPCR reaction without target RNA. Positive controls of RNA were extracted from SGC7901 cells obtained from the Cancer Center of Xiamen University (Xiamen, China). The $2^{-\Delta CT}$ algorithm ($\Delta CT = $ Ct. target − Ct. reference) was employed for data analysis (*Maru et al., 2009*).

## The Kaplan–Meier plotter analysis

The prognostic value of candidate circulating mRNAs was analyzed using the Kaplan–Meier plotter database, an online database containing 54,675 genes on survival based on 1065GC samples with a mean follow-up of 33 months (*Szasz et al., 2016*). Overall survival of patients with high and low expression levels of target genes was displayed using Kaplan–Meier survival curves. Hazard ratios (HR) with 95% confidence intervals (CI), and log-rank *p*-values were also calculated and summarized.

## Functional enrichment network

Gene functional network was performed using gene ontology enrichment (GO) analysis. Enrichment map were created using the Cytoscape (v3.6.0). FDR < 0.05 was considered to be significant.

## Statistical analysis

The Mann–Whitney *U* test was used to compare the expression status of circulating mRNAs in normal and GC groups and calculate the relationship between clinicopathologic characteristics and expression levels of relevant mRNA. Data was shown as median and range. Receiver operating characteristic (ROC) curves and the area under the curve (AUC) was used to identify the diagnosis value of selected mRNA (*Brumback, Pepe & Alonzo, 2006*). Cut-off values were assessed at different sensitivities and specificities and at the maximum Youden's index = (sensitivity + specificity − 1) (*Youden, 1950*). Then, the logistic regression model was performed to obtain a combined ROC curve. GraphPad

Prism 6.0 (GraphPad Soft-ware Inc., La Jolla, CA) and SPSS (version 22.0, IBM SPSS co., USA) was used for these statistical analyses. A two-sided $p$ value of less than 0.05 was defined as statistically significant.

## RESULTS

### Identification of candidate mRNAs from Oncomine database

We compared the mRNA expression level in gastric cancer vs. normal samples obtained from Oncomine database. In total, 304 GC samples including 50 diffuse gastric adenocarcinoma, 21 gastric adenocarcinoma, 113 gastric intestinal type adenocarcinoma, 22 gastric mixed adenocarcinoma, 6 gastrointestinal stromal tumor and 92 other GC and 174 normal controls from 5 selected microarray datasets were analyzed (*Chen et al., 2003*; *Cho et al., 2011*; *Cui et al., 2011*; *D'Errico et al., 2009*; *Wang et al., 2012*). As shown in Fig. 1, expression profiling analysis generated 40 altered mRNA expressions in GC. The top 10 overexpressed and underexpressed genes embodied important genes involved in carcinogenesis and progression as well as several uncharacterized candidates (*collagen type VI alpha 3 chain* (*COL6A3*), *serpin family H member 1* (*SERPINH1*), *pleckstrin homology and RhoGEF domain containing G1* (*PLEKHG1*), *collagen type I alpha 2 chain* (*COL1A2*), *claudin 1* (*CLDN1*), *metallopeptidase inhibitor 1* (*TIMP1*), *NOP56 ribonucleoprotein* (*NOP56*), *collagen type IV alpha 1 chain* (*COL4A1*), *immunoglobulin like domain containing receptor 1* (*ILDR1*); *cadherin 11* (*CDH11*), *pepsinogen 4, group I pepsinogen A* (*PGA4*), *potassium voltage-gated channel subfamily E regulatory subunit 2* (*KCNE2*), *gastric intrinsic factor* (*GIF*), *solute carrier family 9 member A4* (*SLC9A4*), *estrogen related receptor gamma* (*ESRRG*), *ATPase H+/K+ transporting subunit alpha* (*ATP4A*), *ATPase H+/K+ transporting subunit beta* (*ATP4B*), *PRDM16 divergent transcript* (*FLJ42875*), *MAS related GPR family member D* (*MRGPRD*) and *tripartite motif containing 50* (*TRIM50*), respectively).

### Experimental validation of the potential noninvasive biomarkers

To validate our expression profiling analyses, six selected genes including top three overexpressed (*COL6A3*, *SERPINH1* and *PLEKHG1*) and underexpressed (*PGA4*, *KCNE2* and *GIF*) mRNAs were experimental verified using circulating mRNA extracted from 56 GC patients and 14 healthy subjects.

The expression of *COL6A3*, *PLEKHG1*, *PGA4* and *KCNE2* was detected in 55/56, 41/56, 32/56 and 55/56 GC plasma samples respectively, whereas the expression of *SERPINH1* and *GIF* was detected in all GC samples. As shown in Fig. 2, the expression level of *COL6A3* ($P = 0.0128$), *SERPINH1* ($P = 0.0217$) and *PLEKHG1* ($P = 0.0064$) was significantly higher in GC plasmas than those in normal controls. However, the expression levels of *PGA4*, *KCNE2* and *GIF* had no significant change.

### Diagnostic value of plasma *COL6A3*, *SERPINH1* and *PLEKHG1* for GC

We performed ROC curve analysis to assess the diagnostic value of these three circulating mRNAs (Fig. 3). The plasma level of *COL6A3* had a sensitivity of 100% and a specificity of 46.2% to differentiate the GC patients from the healthy subjects, with an AUC of 0.720 at

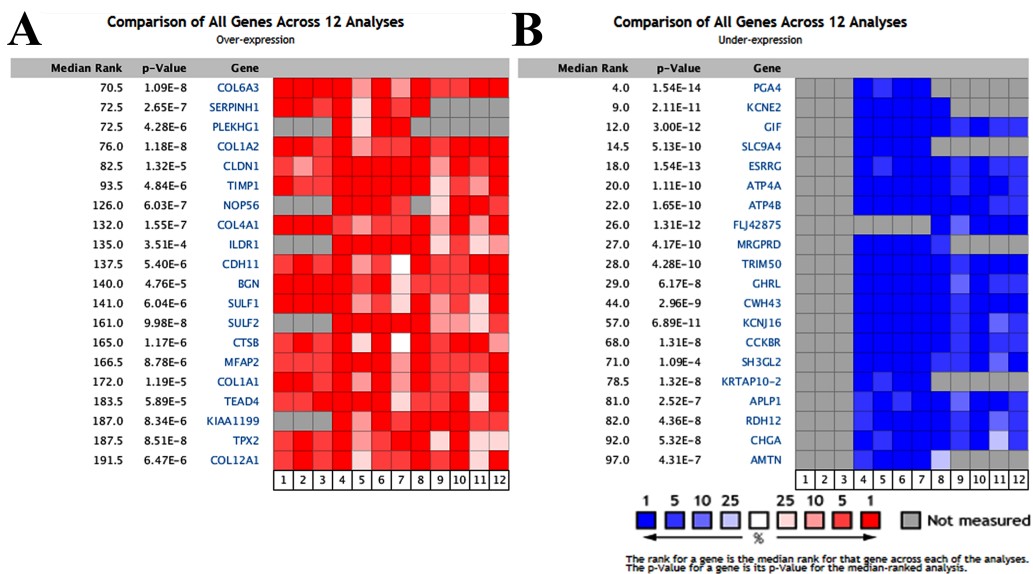

**Figure 1** **Transcriptional heat map of the top 20 over- and underexpressed genes in gastric cancer samples compared with normal samples through Oncomine analysis.** The level plots depict the frequencies (%) of (A) over- and (B) underexpressed candidate messenger RNAs (mRNAs) in 12 analyses from five included studies (*Chen et al., 2003*; *Cho et al., 2011*; *Cui et al., 2011*; *D'Errico et al., 2009*; *Wang et al., 2012*). Red cells represent overexpression. Blue cells represent underexpression. Gray cells represent not measured.

the optimal cut-off point (95% CI [0.522–0.919]). *SERPINH1* had a sensitivity of 58.9% and a specificity of 78.6%, with an AUC of 0.698 (95% CI [0.543–0.852]). *PLEKHG1* had a sensitivity of 68.3% and a specificity of 100%, with an AUC of 0.833 (95% CI [0.699–0.968]). Combining any two of the biomarkers had values for ROC area ranging from 0.676 to 0.870, sensitivities from 60% to 82.9%, specificities from 76.9% to 100% (Fig. 4). Furthermore, we used logistic regression analysis to combine these three circulating mRNAs and obtained an increased AUC value of 0.907 (95% CI [0.820–0.993]), with a sensitivity of 82.9% and a specificity of 100%.

## Associations between clinicopathological characteristics and plasma *COL6A3*, *SERPINH1* and *PLEKHG1*

The associations between clinicopathological characteristics and these three circulating mRNAs were investigated. As shown in Table 2, the higher *COL6A3* level was significantly associated with increased lymph node metastasis ($P = 0.0233$), whereas the elevated expression of *SERPINH1* was associated with advanced age ($P = 0.0034$) and poor differentiation ($P = 0.0231$) in GC patients. No significant association between *PLEKHG1* and any clinicopathological characteristics including age, sex, stage, lymph node metastasis or differentiation was found.

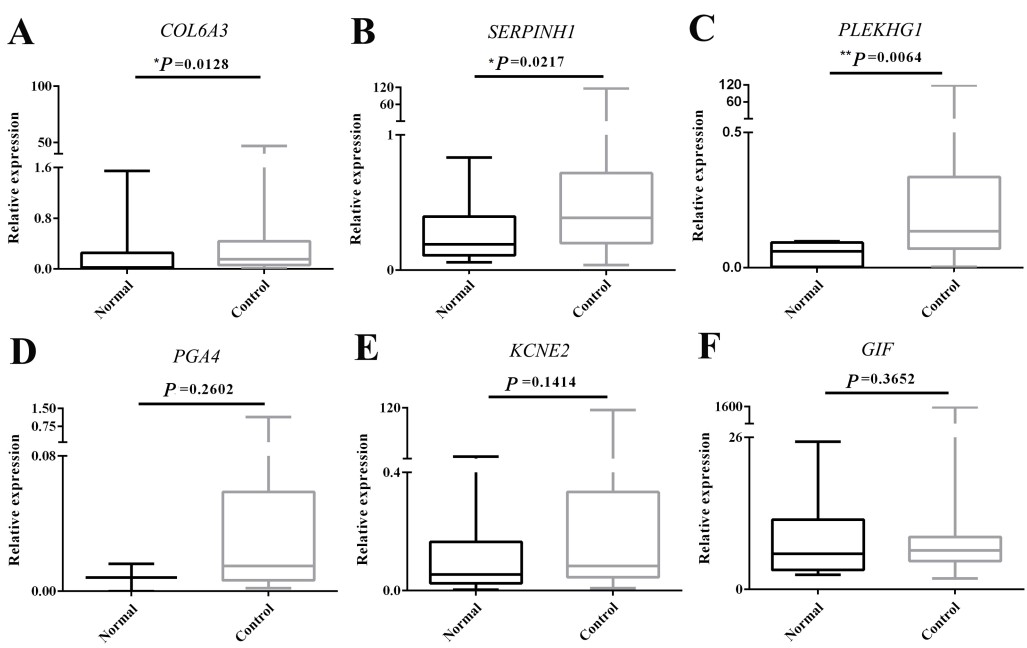

**Figure 2 Experimental validation of the top 6 over- and underexpressed messenger RNAs (mRNAs).**
The change in circulating mRNA levels of *collagen type VI alpha 3 chain (COL6A3)* (A), *serpin family H member 1 (SERPINH1)* (B), *pleckstrin homology and RhoGEF domain containing G1 (PLEKHG1)* (C), *pepsinogen 4, group I (pepsinogen A) (PGA4)* (D), *potassium voltage-gated channel subfamily E regulatory subunit 2 (KCNE2)* (E) and *gastric intrinsic factor (GIF)* (F) between gastric cancer patients and normal subjects detected by qPCR using Student's *t* test. The Mann–Whitney *U* test was used to compare the expression status of circulating mRNAs in normal and GC groups. Data was shown as box plots and the intersecting line represents the median value with the interquartile range. Results were shown with means ± SEM. *$p < 0.05$, **$p < 0.01$.

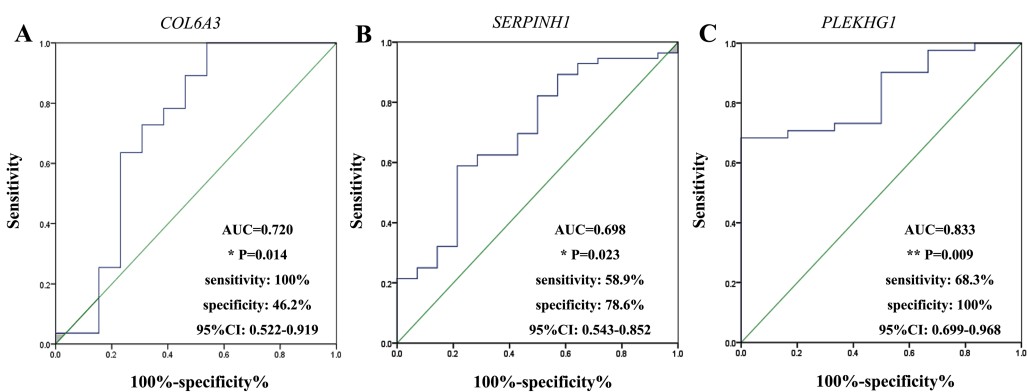

**Figure 3 Receiver-operating characteristic curves (ROC) analysis of selected markers in gastric cancer.** The results showed the performances of fold-change in *collagen type VI alpha 3 chain (COL6A3)* (A), *serpin family H member 1 (SERPINH1)* (B) and *pleckstrin homology and RhoGEF domain containing G1 (PLEKHG1)* (C) messenger RNAs (mRNAs) expression in predicting the gastric cancer. *$p < 0.05$, **$p < 0.01$.

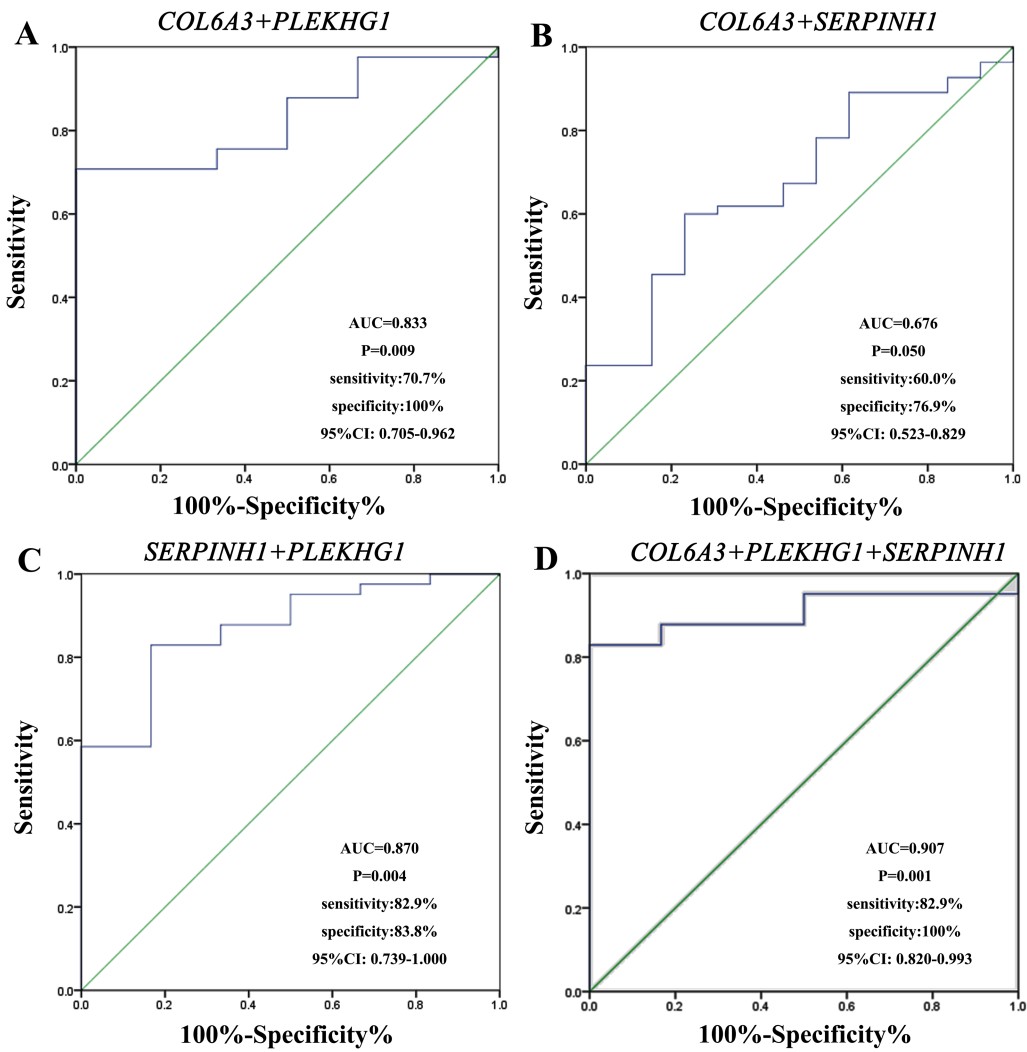

**Figure 4  Receiver-operating characteristic curves (ROC) for combining markers in gastric cancer.** The results showed combination of *collagen type VI alpha 3 chain (COL6A3)* and *pleckstrin homology and RhoGEF domain containing G1 (PLEKHG1)* (A), *COL6A3* and *serpin family H member 1 (SERPINH1)* (B), *SERPINH1* and *PLEKHG1* (C), and combination of these three genes (D) to differentiate patients with gastric from normal subjects.

## Increased *COL6A3*, *SERPINH1* and *PLEKHG1* is associated with poor prognosis

To further evaluate whether the expression levels of *COL6A3*, *SERPINH1* and *PLEKHG1* can predict prognosis, we performed a survival analysis based on publicly gene expression datasets from the Kaplan–Meier Plotter resource. As shown in Fig. 5, the higher expression of *COL6A3* (HR = 1.32, 95% CI [1.11–1.58], $p = 0.0018$), *SERPINH1* (HR = 1.97, 95% CI [1.61–2.41], $p = 3.1e^{-11}$) and *PLEKHG1* (HR = 1.34, 95% CI [1.07–1.69], $p = 0.012$) were all significantly correlated with poor OS in GC. These results indicated that GC patients with high *COL6A3*, SERPINHI or *PLEKHG1* tend to have unfavorable outcome.

Cao et al. (2019), *PeerJ*, DOI 10.7717/peerj.7025

**Table 2** Association between the expression of circulating *COL6A3*, *SERPINH1* and *PLEKHG1* in gastric cancer and clinicopathologic characteristics.

| Variable | COL6A3 | | | SERPINH1 | | | PLEKHG1 | | |
|---|---|---|---|---|---|---|---|---|---|
| | n | Expression status | P | n | Expression status | P | n | Expression status | P |
| Age | | | | | | | | | |
| <50 | 12 | 0.179 (0.028–1.250) | | 12 | 0.190 (0.073–1.430) | | 10 | 0.130 (0.003–1.052) | |
| ≥50 | 43 | 0.147 (0.011–47.014) | 0.7091 | 44 | 0.47 (0.374–116.407) | 0.0034** | 31 | 0.148 (0.036–119.46) | 0.4517 |
| Sex | | | | | | | | | |
| Male | 42 | 0.150 (0.011–10.386) | | 43 | 0.399 (0.374–8.283) | | 30 | 0.126 (0.036–2.447) | |
| Female | 13 | 0.162 (0.018–47.014) | 0.5496 | 13 | 0.325 (0.038–116.41) | 0.1856 | 11 | 0.161 (0.003–119.46) | 0.9633 |
| Stage | | | | | | | | | |
| I + II + III | 36 | 0.150 (0.011–47.014) | | 37 | 0.374 (0.038–116.41) | | 27 | 0.127 (0.003–119.46) | |
| IV | 18 | 0.190 (0.018–10.386) | 0.1616 | 18 | 0.454 (0.118–3.089) | 0.4697 | 13 | 0.198 (0.065–1.214) | 0.5101 |
| Lymph node metastasis | | | | | | | | | |
| <15 | 30 | 0.106 (0.229–47.014) | | 31 | 0.374 (0.221–116.41) | | 20 | 0.118 (0.036–119.46) | |
| ≥15 | 7 | 0.225 (0.155–0.698) | 0.0233* | 7 | 0.472 (0.039–3.623) | 0.6848 | 7 | 0.116 (0.037–0.336) | 0.6207 |
| Differentiation | | | | | | | | | |
| Poor | 19 | 0.198 (0.011–47.01) | | 19 | 0.362 (0.127–116.41) | | 12 | 0.110 (0.003–119.46) | |
| Moderately or well-differentiated | 11 | 0.033 (0.018–0.693) | 0.881 | 11 | 0.678 (0.159–3.089) | 0.0231* | 10 | 0.106 (0.048–2.447) | 0.1059 |

**Notes.**
*P value < 0.05.
**P value < 0.01.

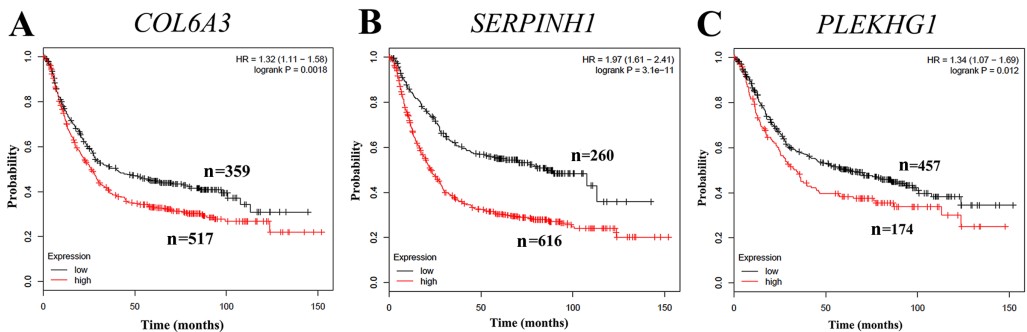

**Figure 5** Correlation of *collagen type VI alpha 3 chain (COL6A3), serpin family H member 1 (SER-PINH1)* and *pleckstrin homology and RhoGEF domain containing G1 (PLEKHG1)* with survival outcomes in gastric cancer patients. Increased expression of (A) *COL6A3*, (B) *SERPINH1* and (C) *PLEKHG1* predicted worse overall survival (OS) in gastric cancer.

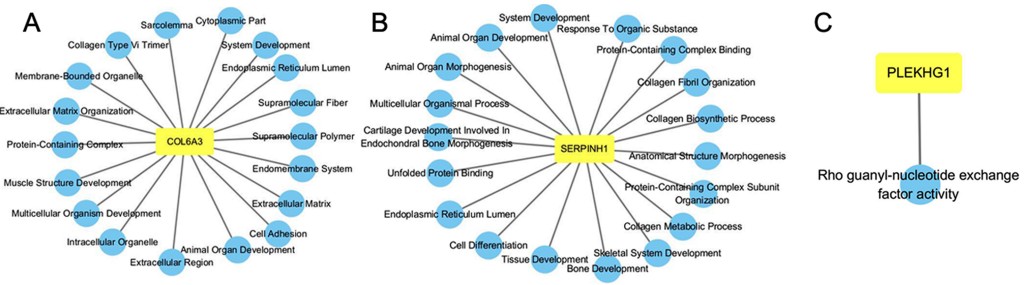

**Figure 6** GO functional enrichment analysis of *COL6A3* (A), *SERPINH1* (B) and *PLEKHG1* (C).

## GO functional enrichment analysis

Functional enrichment network of *COL6A3*, *SERPINH1* and *PLEKHG1* was constructed. As shown in Fig. 6A, *COL6A3* was predicted to have the main functions: extracellular matrix organization, cell adhesion, multicellular organism development, animal organ development and system development. *SERPINH1* played roles in extracellular matrix organization, collagen fibril organization, skeletal system development, collagen metabolic process, and animal organ morphogenesis (Fig. 6B). *PLEKHG1* was found associated with Rho guanyl-nucleotide exchange factor activity (Fig. 6C).

## DISCUSSION

The survival of GC affected patients depends mainly on early detection (*Wang et al., 2006*). The common screening approaches are gastroscopy and computed tomography, which are invasive and expensive (*Ke et al., 2017*). Therefore, easily accessible and noninvasive biomarkers derived from body fluid are prevalent (*Shen et al., 2017*). Exploration of circulating biomarkers for various cancer types can be conducted through different approaches. In the present study, we identified candidate biomarkers according to our previous strategy which combined comprehensive analysis of microarray data and

experimental validation using plasma samples (*Zhou et al., 2017*). We first identified three circulating mRNA markers (*COL6A3*, *SERPINH1* and *PLEKHG1*) that carry diagnostic potential for GC. We also found that the combination of these three markers exhibited better diagnostic performance for GC than each individual marker. Our strategy can also be flexibly applied to various diseases.

Circulating RNA including lncRNA, mRNA and microRNA can be isolated and detected in serum, plasma, urine and lymph. In bodily fluid, RNA molecules are directly exposed to RNase, resulting in the degradation of RNAs and the difficulty to identify RNA based biomarkers (*Hasselmann et al., 2001*; *Sisco, 2001*). However, some studies have suggested that circulating RNA is especially stable due to the protection from phospholipids (*Elhefnawy et al., 2004*; *Halicka, Bedner & Darzynkiewicz, 2000*; *Ma, Tao & Kang, 2012*). In addition, another study demonstrates that the concentration of circulating RNA in GC patients is higher than that of healthy controls, which is associated with tumor growth and metastasis metabolism (*Elhefnawy et al., 2004*; *Rykova et al., 2006*). In this study, we found that the levels of plasmatic *COL6A3*, *SERPINH1* and *PLEKHG1* were significantly increased in patients with GC than in healthy subjects.

*COL6A3*, located at chromosome 2q37, encodes the alpha-3 chain for type VI collagen (*Dankel et al., 2014*). Collagen VI has been initially defined as an extracellular matrix protein and it is expressed in various tissues such as muscle, skin and cartilage. *COL6A3* is a secreted protein and have received growing attention due to its abnormal expression in colon, pancreatic, bladder and prostate cancer (*Kang et al., 2014*; *Thorsen et al., 2008*). In a previous study, *COL6A3* has been shown to be a potential plasma marker of colorectal cancer and is associated with tumor metastasis (*Qiao et al., 2015*). However, the expression pattern and function of *COL6A3* in the tumorigenesis of GC remain unclear. Our present study indicated that *COL6A3* was overexpressed in plasma of GC patients and was associated with increased lymph node metastasis.

*SERPINH1*, also known as heat shock protein 47 (HSP47), belongs to the serpin superfamily involving serine proteinase inhibitors (*Ito & Nagata, 2016*). The location of *SERPINH1* is at chromosome 11q13.5, a domain frequently abnormal in various human cancers. Numerous studies have demonstrated that *SERPINH1* is overexpressed in various human cancers, including lung cancer, pancreatic cancer, cervical cancer and glioma (*Wu et al., 2016*; *Yamamoto et al., 2013*). In addition, serum *SERPINH1* has been reported to be used as a possible target for patients with scirrhous gastric cancer. Our results showed that the overexpressed *SERPINH1* was associated with advanced age and poor differentiation in plasma from GC patients. These data indicated that *SERPINH1* played a critical role in GC, although further studies are necessary to clarify the biological mechanism of *SERPINH1* in GC.

*PLEKHG1* contains a Rho guanine nucleotide exchange factor domain and a pleckstrin homology domain. *PLEKHG1* acts as a signaling platform in various cells, but the detail functions are not clear. In this study, we found that the plasma level of *PLEKHG1* mRNA was significantly increased in GC patients compared with normal subjects, with a markedly high AUC value of 0.8333. These results suggested that *PLEKHG1* mRNA have a high diagnosis capability.

We carried out the ROC curve to analyze the diagnostic value of *COL6A3*, *SERPINH1* and *PLEKHG1* in plasma from GC patients. The results demonstrated that *PLEKHG1* had higher diagnostic value for GC than that of *COL6A3* or *SERPINH1*. More powerful diagnostic values were observed when combining these three mRNAs, resulting in an AUC of 0.907. In addition, the prognostic roles of these three potential biomarkers in GC patients were rarely reported and all these biomarkers were correlated with worse OS for patients with GC.

## CONCLUSIONS

Accordingly, we have identified potential noninvasive biomarkers for gastric cancer using bioinformatics analysis through a public database and verified their value using GC clinical tumor and plasma specimens. *COL6A3*, *SERPINH1* and *PLEKHG1* are three prospective biomarkers for GC. The combination of plasma *COL6A3*, *SERPINH1* and *PLEKHG1* represent a promising diagnostic method. The clinical samples employed in this study were relatively limited. Hence, large-scale studies should be performed to investigate the clinical significance of COL6A3, SERPINH1 and PLEKHG1 in GC in the future. Moreover, further investigation of their biological function and their potential as therapeutic targets in GC is warranted.

## ACKNOWLEDGEMENTS

We would like to thank the patients and healthy subjects for consenting to provide material for this study.

### Funding

This study was supported by the National Natural Science Foundation of China (No. 81802323 and No. 81702414), Natural Science Foundation of Fujian Province of China (No. 2017Y0084 and No. 2015J01557), Science and Technology Service Network Initiative Foundation of CAS (No. 2016T3009), the Fujian Provincial Health and Family Planning Commission Foundation of Youth Scientific Research Project (No. 2015-2-43), and Xiamen Science and Technology Bureau Foundation of Science and Technology Project for the Benefit of the People (No. 3502Z20164010). The funders had no role in study design, data collection and analysis, decision to publish, or preparation of the manuscript.

### Grant Disclosures

The following grant information was disclosed by the authors:
National Natural Science Foundation of China: 81802323, 81702414.
Natural Science Foundation of Fujian Province of China: 2017Y0084, 2015J01557.
Science and Technology Service Network Initiative Foundation of CAS: 2016T3009.
Fujian Provincial Health and Family Planning Commission Foundation of Youth Scientific Research Project: 2015-2-43.

Xiamen Science and Technology Bureau Foundation of Science and Technology Project for the Benefit of the People: 3502Z20164010.

## Competing Interests

The authors declare there are no competing interests.

## Author Contributions

- Wei Cao performed the experiments, analyzed the data, prepared figures and/or tables.
- Dan Zhou conceived and designed the experiments.
- Weiwei Tang contributed reagents/materials/analysis tools.
- Hanxiang An and Yun Zhang authored or reviewed drafts of the paper, approved the final draft.

## Human Ethics

The following information was supplied relating to ethical approvals (i.e., approving body and any reference numbers):

All experimental protocols were approved by the Clinical Research Ethics Committee of the First Affiliated Hospital of Xiamen University. All methods were performed in accordance with the Declaration of Helsinki. Written informed consent was obtained from all human participants after complete description of the study.

## Data Availability

The raw data is available as Dataset S1.

## Supplemental Information

Supplemental information for this article can be found online at http://dx.doi.org/10.7717/peerj.7025#supplemental-information.

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
