# Peer review of "Discovery of plasma messenger RNA as novel biomarker for gastric cancer identified through bioinformatics analysis and clinical validation"

_PeerJ, doi:10.7717/peerj.7025_

## Round 0.1 · original submission · Major Revisions

The reviewers would like to see some revisions made to your manuscript. I invite you to respond to the reviewers' comments and revise your manuscript. I hope that you find the comments of the reviewers useful. If you decide to revise the work, please send a point-by-point response to the reviewers comments, and a revised manuscript with changes made visible. Furthermore, in the process, you can ask a native speaker for help.

·

Basic reporting

The article explored further putative biomarkers for gastric cancer disease. The article is well organized and comprehensive. However, the following notes need to be adjusted.

Abstract
Results: rephrase the results of association with clinicopathological features to be more concise with clear language.
Conclusion: were >>> could be putative

Experimental design

Methodology
304 GC cancer: different number of samples are noted in the Figures
Line 84: available from each study >>> available for each study

Validity of the findings

Results
Could you explain how you combine ROC curve by log regression analysis? To have at the end a value higher than a single value and not the mean of the three values?



Figures
Unify font size of p values in Fig. 2
Fig 2 is box plot not scatter plot
Set cutoff values, sensitivity and specificity in Fig. 3
Unify format of ROC in Fig 3 and 4
Write more labels on Fig. 4

Additional comments

Discussion
Further addition enrichment analysis to identify putative function of the three significant genes is warranted.

Reviewer 2 ·

Basic reporting

The paper is easy to follow, not complicated, even pedagogical. In few words, it is even too simple and it is not simple for the reader to know if she/he believed in the results or she/he thank that she/he believed in the results. Some parts must be more detailed to have a better view of the experiment.

Experimental design

it is properly done in the way that it is explained, but there is a break with the results. Reader expect a large number of complicated cases… and it is directly restricted. Similarly, it is not clear how the different types of adenocarcinoma 
are treated. Is the merging of adenocarcinoma 
correct ?

Validity of the findings

As said before, it is easy to follow, but how is done the reduction of the selected genes? How is it sensitive? More details must be provided. And more analyses must be done. Is their some co-regulation? Some proteins implicated in similar pathways, same pathways. In few words, it is a bit too short. For instance, some p-values are not so high, aren’t they?
As said in PeerJ ‘Speculation is welcome, but should be identified as such.’

Additional comments

I like the idea and design, but cannot say if I think/believe it is properly done at this level.

Reviewer 3 ·

Basic reporting

Overall, the study has merit and the technical approach is valid, however, It appears that this manuscript is not carefully revised. The manuscript requires improvement in several concerns which will be written in details in the comments for the authors' section.

Experimental design

- The comments will be written in details in the comments for the authors' section.

Validity of the findings

- The comments will be written in details in the comments for the authors' section.

Additional comments

The manuscript requires improvement in the following concerns:

- The authors are advised to copyedit the manuscript: a native English speaking colleague could help them with this issue, if possible, or they may need to use a professional language editing service.

- Authors are advised to revise all the manuscript to ensure that all abbreviations have been written in their long name at their first mention in the text, tables, and figures, especially the gene names.

-Authors are advised to revise all the manuscript to ensure that all gene names are written in italic font, especially (only for example) lines: 147-149, in the tables and figures.

Abstract:
- Lines 38, 39: The authors mentioned that “expression levels of COL6A3, SERPINH1 and PLEKHG1 were significantly downregulated in GC plasmas” Are these genes upregulated or downregulated?

Methods:
- Why did the authors select this specific type of database? It is better to give the readers a concise hint about its advantages relative to other online databases.

- The characteristics of the dataset used in the current study should be presented in a table for better clarification and readability (even if supplied as suppl. materials)

- The authors did not mention how they selected the study data and the outcomes of the database, the filtration process, the quality control measures they follow in their work on Oncomine database to facilitate the replication of their work by interested future researchers.

- The authors validated their findings on 56 GC patients and 14 healthy subjects. On what basis they selected their controls (inclusion and exclusion criteria) and how they knew they are healthy subjects?

- What is the reference they follow or their institute follows to diagnose gastric cancer? This answer will help readers to some extent to follow the authors and understand the clinicopathological presentation/association tables (1 and 2).

- It was not clear why did the authors did two consecutive centrifugation steps at two different speed for the collected samples. Please clarify.

- Please add the characteristics of the included controls to Table (1). Why did the authors do the RT in triplicate meanwhile they did the qPCR in duplicate? In other words what was the aim of the former biological replicates?, each RNA sample yield 3 tubes included cDNA then what the further processing steps with these 3 replicates. Please clarify to the readers.

- Thanks to the authors for providing the primer sequences in a suppl. table. Are these primers self-designed or derived from other published work? If the former, please provide the name of the software you applied in its formal citation in the text, or provide the citations you follow if they were derived from other publications.

- I advise the authors to read and refer to the MIQE (minimum information for publication of quantitative real-time PCR experiments) guidelines ( PMID: 19246619) for reporting their qPCR methodology. Although they explain the extraction step in details, the methodology of their quantitative PCR needs more elaboration about the type of the syber green mix they used the amount/concentrations they applied for the samples before mention of the PCR programming.

- In suppl. table for the primers sequences, the melting temp. for the primers was 59°C, why did the authors run their PCR at 58°C?

Statistical analysis:
- Please update your reference of the SPSS program as it is related to “IBM SPSS” co.

References:
- Lines 337, 364: Please provide the volume/page range and/or the DOI of the specified references.

Figures and Tables
- Each figure and table should be self-exploratory; all mentioned abbreviations either included on the figure/table or its legend should be written in details in the legend.
- In addition to the above related general comment, more specifically:
A- Figure 1: Are the P values mentioned in the heat Map corrected ones? What about the false discovery rate, why did not the authors apply?
B- Figure 2: please mention the statistical test applied for this comparison.
C- Figure 3: the authors should correct the gene name of (C) as it is PLEKHG1 not GPT2, list the abbreviations mentioned in the figure as recommended above and indicate the value at which the P is significant.
D- Table 1: The authors should add the control group characteristics.
What is meant by (59±26) in the 2nd raw?
Please add in the table footnote how are the data presented? The stage of cancer classified on which basis (provide the reference the authors followed), Why did the authors apply the value of 15 for lymph node metastasis classification?
Please revise the total of the stage (I+II+III: 37) + (IV: 18) = 55 and the total GC patients wear 56?!
The number of L.N metastasis (31+7) involved which stages? Please clarify for the readers.
Why did according to the differentiation classification of the patients, the total are (19+11) = 30?
Please all these issues should be clarified in the table footnote

E- Table 2: it needs an extensive revision
- Add “circulating” after “expression of” in the table title
- Add to the footnote the type of data presented in the table, the abbreviations, the type of statistical test applied and what “*” and “**” did mean?
- Add the total number (n = ) for each column to facilitate following the results.
- Why did for each gene the total of stages (I+II+III) and (IV) is less than the total number of patients by one?
- Please replace the terminology “under moderately” by an appropriate one. Pay attention to be consistent for the applied terminology all over the manuscript to avoid confusion of the readers.

Discussion:
It is concise and related to the study findings, just the authors should provide the study limitation(s) by the end of the discussion.

Minor comments:
- Line 60: the terminology “recently” is not appropriate as the cited reference was 2015.
- Line 82: p-value≤10-4 let the “-4” superscript.
- Line 113: –ΔCT should be superscript.
- Line 149: add “, respectively” after the closing bracket.

Reviewer 4 ·

Basic reporting

English requires general adjustments (e.g. inappropriate use of definite articles).

Experimental design

Given the heterogeneity of gastric cancer, sub-analyses have to be provided for different biological and histopathological entities. In general, it does not appear to make sense to include mesenchymal tumours like GIST and gastric tumours "other" that could possibly be also lymphoma or carcinoid, i.e. non-carcinomatous entities in the same analyses.

Fig. 1: It is unclear what "12 analyses from 5 included studies" really means. What kind of studies? What kind of analyses? As it stands, it is not possible to understand what these data really mean and which cancer subtypes were included. Furthermore, why are all cells in Fig. 1B for analyses 1-3 grey?
A comprehensive data file on the primary analyses should be provided as a supplement.

The type of GC (histological/biological sub-type) of the patients included are not stated. Furthermore, the ratio of cases to samples is skewed about 4-fold towards cases. It would make statistically a lot more sense to reverse this ratio, i.e. include around 250 controls. At least, a 1-to-1 ratio has to be provided before results can be trusted. This is also important to interpret the ROC analyses.

Validity of the findings

Crucially, no survival analysis nor mRNA data on the tumour specimens has been provided for the patient cohort investigated. Hence, it is completely unclear whether plasma mRNA and cancer mRNA levels show any relevant correlation.

Costs for radiological and endoscopic procedures are NO argument for PCR-based diagnostics of gastric cancer. These techniques have to be performed if GC is suspected. Furthermore, given the sensitivity and specificity results, the suggested qPCR-based biomarkers are likely not suited for a screening of GC - this has to be discussed. Furthermore, it is not stated what the possible purpose of the biomarker would be - secondary prevention? Biological dignity? Diagnostic?

Reviewer 5 ·

Basic reporting

Nothing

Experimental design

Nothing

Validity of the findings

Nothing

Additional comments

This study identified three serum mRNAs, COL6A3, SERPINH1 and PLEKHG1, as unfavarite prognostic factors in gastric cancer, using Oncomine database. The authors evaluated the usefulness of the three mRNA, using 56 gastric cancer patients and 14 healthy subjects. The authors concluded that these three mRNA are useful biomarker in gastric cancer patients. This study may be a promising. However, there are some drawbacks.

Major1. The study compared serum of gastric cancer patients with healthy subjects, but did not compare gastric cancer with other cancers such as colon cancers, pancreatic cancer, and lung cancers. That is why the specificity of these biomarkers for gastric cancers may be relatively low. These biomarkers are specific for gastric cancer?

Major 2. Are there any differences in outcomes between gastric cancer patients with positivity of three biomarkers and with positivity of one or two biomarkers.

Major 3. Table 1. The cut off levels of lymphatic metastasis is 15. How was the cut of level of lymphatic metastasis determined? Is there any difference in outcomes of patients?

Minor1. In Abstract, line 38-39, PLEKHG1 were significantly downregulated……
Is this description true? Probably, upregulated?

---

## Round 0.2 · accepted · Accept

I think this manuscript is good enough now.

# Reviewer 2 ·

Basic reporting

Authors have properly answered my questions.

Experimental design

Authors have explained more precisely their hypotheses.

Validity of the findings

It is a scientific paper with a correct design.

Additional comments

Authors have properly answered my questions.

Reviewer 3 ·

Basic reporting

All the required concerns have been done.

Experimental design

All the required concerns have been done.

Validity of the findings

All the required concerns have been done.

Additional comments

The authors have responded to all concerns raised previously by the referee. Thanks

Reviewer 4 ·

Basic reporting

I will only respond to the rebuttal concerning the issues I raised with the previous version:

Effort was made to improve language.

Experimental design

I will only respond to the rebuttal concerning the issues I raised with the previous version:

The authors did not care to take out non-adenocarcinoma of in silico analyses. Why?

Validity of the findings

I will only respond to the rebuttal concerning the issues I raised with the previous version:

I was not suggesting to increase the case number - but the control number, which should easily be doable. The reluctance of the authors to provide any further experimental data does not seem plausible. Do they consider a power below 90% to be sufficient for biomarker discovery studies?

The authors did not add data on mRNA expression of their biomarker candidates from primary cancer tissue - which has to be provided if claiming any relevance of biomarker.

Additional comments

I will only respond to the rebuttal concerning the issues I raised with the previous version:

The authors still vastly overstate the potential role of a qPCR-based biomarker approach. Their gross misjudgement of the role of endoscopy for evaluation of gastric cancer casts serious doubts on their understanding of the role of biomarkers for the diagnosis of cancer. They were given the opportunity to elaborate on this - but did not do so. In particular, no considerations of false-positive results and what to possibly do with these are made. This kind of scientific reasoning is dangerous and might have detrimental effects on health care.